# Common attributes in retired professional cricketers that may enhance or hinder quality of life after retirement: a qualitative study

Stephanie R Filbay,[1] Felicity Bishop,[2] Nicholas Peirce,[3] Mary E Jones,[1] Nigel K Arden[1]

► Prepublication history and additional material are available. To view these files please visit the journal online (http://dx.doi.org/10.1136/bmjopen-2017-016541).

[1]Arthritis Research UK Centre for Sport, Exercise and Osteoarthritis; Nuffield Department of Orthopaedics, Rheumatology and Musculoskeletal Sciences, University of Oxford, Oxford, UK
[2]Department of Psychology, University of Southampton, Southampton, UK
[3]National Centre for Sport and Exercise Medicine and National Cricket Performance Centre, Loughborough University, Leicester, UK

**Correspondence to**
Dr Stephanie R Filbay;
stephanie.filbay@uq.net.au

## ABSTRACT

**Objectives** Retired professional cricketers shared unique experiences and may possess specific psychological attributes with potential to influence quality of life (QOL). Additionally, pain and osteoarthritis can be common in retired athletes which may negatively impact QOL. However, QOL in retired athletes is poorly understood. This study explores the following questions from the personal perspective of retired cricketers: How do retired cricketers perceive and experience musculoskeletal pain and function in daily life? Are there any psychological attributes that might enhance or hinder retired cricketers' QOL?

**Design** A qualitative study using semistructured interviews, which were subject to inductive, thematic analysis. A data-driven, iterative approach to data coding was employed.

**Setting** All participants had lived and played professional cricket in the UK and were living in the UK or abroad at the time of interview.

**Participants** Eighteen male participants, aged a mean 57±11 (range 34–77) years had played professional cricket for a mean 12±7 seasons and had been retired from professional cricket on average 23±9 years.

**Results** Fifteen participants reported pain or joint difficulties and all but one was satisfied with their QOL. Most retired cricketers reflected on experiences during their cricket career that may be associated with the psychological attributes that these individuals shared, including resilience and a positive attitude. Additional attributes included a high sense of body awareness, an ability to self-manage pain and adapt lifestyle choices to accommodate physical limitations. Participants felt fortunate and proud to have played professional cricket, which may have further contributed to the high QOL in this group of retired cricketers.

**Conclusions** Most retired cricketers in this study were living with pain or joint difficulties. Despite this, all but one was satisfied or very satisfied with their QOL. This may be partly explained by the positive psychological attributes that these retired cricketers shared.

## INTRODUCTION

Research surrounding the long-term outcomes of a professional sporting career has largely focused on negative impacts

## Strengths and limitations of this study

► A strength of this study was the strong themes and consistent patterns that emerged through the interviews and the deep reflection and open, personal experiences that retired cricketers shared that enriched the findings of this study.

► Since participants had been retired from professional cricket for an average of 23 years, we gained unique insight into longer-term quality of life after professional sport which had not been previously explored.

► It is not known whether cricket participation contributed to the development of positive psychological attributes or whether these largely pre-existed prior to cricket participation or developed after retirement.

► Limitations include generalisability and the possibility of selection bias, whereby retired cricketers with specific characteristics may be more likely to desire participation in a qualitative interview study.

rather than investigating the potential for positive psychological impacts that persist beyond retirement.[1] On retirement, athletes commonly experience negative emotions and reduced life satisfaction while transitioning and adapting to an alternative lifestyle,[2] but very little is known about life after this transition. After retirement, athletes are at increased risk of developing osteoarthritis, predominately due to a history of sport-related injury, and this can result in pain and activity limitations.[3–7] However, retired athletes may possess psychological attributes that could enhance quality of life (QOL) and facilitate effective coping with pain and musculoskeletal impairment. Additionally, retired athletes may feel proud and accomplished to have played professional sport which could have positive impacts on QOL. On the other hand, functional limitations in individuals

with high athletic identity could negatively impact QOL if their physical activity desires are not satisfied. These complex possible influences on the QOL of retired athletes are poorly understood.

Professional cricket is all-encompassing, the duration of play exceeds most other sports with formats including entire day matches and test matches lasting up to five consecutive days. In cricket, different playing positions have distinct psychological demands. A batsman facing a ball is under great pressure to perform, one slip of concentration or an incorrect decision can see him out of the game with no further opportunity to contribute positively to the team. A fielder in a slip position maintains prolonged intense focus in preparation for a potential catch with an outcome of success or failure determined by a momentary reaction. The demands of cricket contrast most other team sports where players have greater opportunity to correct performance and ongoing involvement within a game. Thus, professional cricketers face unique competitive demands that require advanced psychological and behavioural control to manage stress and perform under pressure.[8 9] The ability to deal with such stressors will, in part, determine the success of a professional cricketer. Psychological attributes associated with a successful cricket career include resilience, mental toughness, self-belief, optimism, coping with adversity and confidence.[10–12] Although these psychological attributes have been observed in professional cricketers, it is not known whether they persist after retirement. The psychological attributes that retired cricketers possess have potential to benefit other aspects of life and reduce the impact of musculoskeletal pain following retirement from cricket.

We undertook a qualitative study to provide an in-depth analysis of life after retirement from professional cricket, from the personal perspective of retired cricketers. The a priori aim of this qualitative study was to explore physical activity after retirement from professional cricket. However, the study also captured participants' broader perspectives regarding QOL and prominent themes linking QOL to professional cricket emerged during the initial inductive analysis. This paper focuses on these emergent themes, which were explored while considering two sensitising questions: How do retired cricketers perceive and experience musculoskeletal pain and function in daily life? Are there any psychological attributes that might enhance or hinder retired cricketers' QOL? A complementary manuscript will address physical activity after retirement from professional cricket.

## METHODS

The study methodology and results are reported in accordance with the consolidated criteria for reporting qualitative research guidelines.[13]

## ETHICAL CONSIDERATIONS

This study has been approved by Medical Sciences Inter-divisional Research Ethics Committee (IDREC), University of Oxford (reference number R45197/RE001).

## PARTICIPANTS AND RECRUITMENT

Participants were purposively sampled from a larger cohort study of 187 retired professional English cricketers who recently completed a cross-sectional questionnaire collecting information on demographics, playing history, current pain, medical history (including doctor-diagnosed osteoarthritis, arthroscopic surgery and joint replacement surgery) and injury history (Jones *et al, unpublished data*). In order to facilitate a balanced exploration of physical activity behaviours, barriers and facilitators, we purposively sampled individuals based on their responses to two questions from the cross-sectional study assessing perceived impact of previous cricket participation on current physical activity levels: (1) strongly agree or agree that participation in cricket has resulted in an *increase* in current physical activity level (n=46, 42%) or (2) strongly agree or agree that participation in cricket has resulted in a *decrease* in current physical activity level (n=27, 25%). We aimed to include an equal number of participants from each of these groups.

Only individuals from the larger cross-sectional study who indicated a willingness to participate in future sport-related research were invited (by email) into the current study. The invitation email included an information sheet describing the study purpose, procedure (including data storage and assurance of confidentiality), dissemination plan and the interviewer's credentials. Among those expressing an interest in this study, participants were then purposively sampled to ensure men of a range of ages were interviewed. Interviews were conducted until the point of data saturation, defined a priori as the point at which no new themes were identified from four consecutive interviews (two from participants with reduced physical activity and two from participants with increased physical activity). After which, additional interviews were performed as deemed appropriate to explore ideas and themes in greater detail after following the semistructured interview guide. These final interviews were also used to affirm data saturation. If new themes emerged from these final interviews, additional interviews would be conducted until data saturation is satisfied.

A total of 42 email invitations were sent to eligible participants, 19 emails received no response, two individuals declined to participate, two individuals were unavailable due to overseas travel and one individual indicated a willingness to participate but did not respond to further correspondence. Data saturation was achieved by the 14th interview. An additional four interviews were carried out after data saturation and no new themes emerged. Data from all 18 interviews were used for analysis.

## INTERVIEWS

Audio-recorded semistructured telephone interviews were performed by SRF (a female postdoctoral researcher and physiotherapist with experience in qualitative research). The interviewer was unknown to participants prior to the interview. Informed verbal consent was obtained from each participant prior to taking part. Interviews were transcribed verbatim by a research assistant. All transcripts were de-identified during transcription, and an alias was assigned to each participant. The 18 interviews were on average 26 min in length (ranging from 18 to 37 min).

A semistructured interview guide was used to assure that key areas of interest were covered while enabling the researcher to adapt the interview guide to elicit relevant and rich information from respondents by probing and prompting for additional details.[14] The interview guide was pilot tested with three individuals with cricket experience prior to gaining ethics approval. The interview guide contained open-ended questions to provide participants with the greatest opportunity to consider personal perspectives and experiences (see online supplementary appendix 1). The interview guide was iteratively adapted throughout the interviews to incorporate any additional issues of importance to respondents (eg, by adding a question to explore their relationship with cricket post-retirement). Interviewees were given the opportunity to contribute any additional information at the end of the interview.

## ANALYSIS PROCEDURE

An inductive thematic approach was used[15 16] facilitated by NVivo V.11 software.[17] Following each interview, the interviewer summarised and reflected on initial ideas in a study journal. Transcripts were read and re-read with accompanying audio to identify all information relevant to the research aims.[18] An iterative approach to data coding was employed. All data that might be relevant to the aims of the study were coded into multiple categories to be refined and analysed for themes in subsequent stages of analysis.[15] During the earlier stages of analysis, no attention was given to the prevalence of themes and coding was performed diversely and inclusively, withholding further interpretation. In line with an inductive thematic approach, coding was data-driven, performed without reference to a pre-existing coding structure and with little engagement with literature to avoid sensitisation to specific themes.[15 18]

During the later stages of analysis, the data were further analysed for repeated patterns, codes were sorted into a hierarchical structure representing themes and subthemes, overlapping themes were merged and those outside the scope of the current study were filed separately. These themes and subthemes were repeatedly reviewed and refined to confirm external heterogeneity and internal homogeneity within themes and to ensure an accurate representation of the entire dataset.[15 19] During this stage, all transcripts were re-read, and the study journal was revisited to check that themes accurately reflected the key issues discussed by participants. A selection of six transcripts representing participants with diverse physical activity patterns were analysed by a second investigator (FLB) blinded to the coding structure developed by the first author (SRF). A meeting was then held to discuss key themes, and a high level of agreement was achieved between investigators. Key themes and subthemes will be described with reference to participant quotes.[15 19]

Following qualitative analysis, responses to relevant questions collected as part of the cross-sectional study were used to describe the study sample (including osteoarthritis status, age, previous surgery and length of professional cricket career). Questionnaire data were cross-checked for accuracy with participants' narratives and if disparities were present (eg, a participant reported no current joint pain on questionnaire but described experiencing current joint pain during the interview), participants' narratives were considered more reliable.

## RESULTS

### Participants' characteristics

The 18 participants who took part in the interviews were all male, aged a mean 57±11 (range 34–77) years, played professional cricket in the UK for a mean 12±7 (range 2–25) seasons and had been retired from professional cricket for an average 23±9 (range 7–38) years. The majority were overweight by WHO criteria[20] (n=12, 67%), and three participants (17%) were classified as obese. Current joint pain or difficulties were common (n=15, 83%), 10 people (56%) had been told by a doctor that they had osteoarthritis, 12 participants (67%) had received at least one joint surgery and five participants (28%) had received one or more total joint replacement. More than two-thirds (n=14, 78%) reported having ever had an injury that resulted in more than 1 month of reduced participation in exercise, training or sport. Full participant characteristics are reported in table 1.

### Pain and musculoskeletal function

Fifteen retired cricketers described experiencing current pain and limitations with musculoskeletal function. For most individuals, this prevented them from taking part in specific activities, like running and recreational sports. Although this was described as frustrating by some, participants had made lifestyle modifications to accommodate these impairments and very few described a negative impact on QOL.

*Jim: "I've had 13 operations over the years because of cricket injuries and resulting stuff from that and I'm just restricted in what I can do really…it doesn't make me feel, nothing really, as long as I can do the little bits I can I'm quite happy."*

*Sam: "So there are little bits that frustrate me but I mean, I'm nearly 60 so you know, your body does get worn out if you do something as extreme as play a professional sport for*

**Table 1**  Participant characteristics

| Alias | Age range* | BMI | Joint pain | OA | Past injury | Arthroscopic joint surgery | Joint replacement | Years postretirement* | UK professional seasons* | Considering the benefits and risks of cricket, I would do the same again | I would recommend cricket to significant others |
|---|---|---|---|---|---|---|---|---|---|---|---|
| Cam | 51–55 | Overweight | Y | N | Y | N | N | 26–30 | 1–5 | Strongly agree | Strongly agree |
| Dan | 56–60 | Normal | Y | Y | Y | Y | N | 26–30 | 6–10 | Strongly agree | Strongly agree |
| Dom | 61–65 | Obese | N | Y | N | Y | Y | 26–30 | 16–20 | Strongly agree | Strongly agree |
| Gus | 56–60 | Overweight | N | N | N | N | N | 11–15 | 1–5 | Strongly agree | Strongly agree |
| Joe | 61–65 | Overweight | N | N | Y | N | N | 31–35 | 16–20 | Agree | Agree |
| Leo | 76–80 | Normal | Y | Y | Y | Y | Y | 36–40 | 1–5 | Strongly agree | Strongly agree |
| Ned | 56–60 | Overweight | Y | N | N | N | N | 16–20 | 16–20 | Strongly agree | Strongly agree |
| Tim | 36–40 | Overweight | Y | Y | Y | Y | N | 6–10 | NR | Strongly agree | Strongly agree |
| Wes | 66–70 | Overweight | Y | N | Y | N | N | 26–30 | 21–25 | Strongly agree | Strongly agree |
| Ben | 56–60 | Overweight | Y | Y | Y | Y | Y | 21–25 | 11–15 | Strongly agree | Strongly agree |
| Fin | 31–35 | Overweight | Y | N | Y | Y | N | 6–10 | 6–10 | Strongly agree | Agree |
| Guy | 46–50 | Obese | Y | N | Y | Y | N | 21–25 | 1–5 | Strongly agree | Strongly agree |
| Jim | 66–70 | Overweight | Y | Y | Y | Y | Y | 21–25 | 21–25 | Strongly agree | Strongly agree |
| Ken | 56–60 | Overweight | Y | Y | Y | N | N | 26–30 | 6–10 | Strongly agree | Strongly agree |
| Lee | 46–50 | Overweight | Y | N | N | Y | N | 11–15 | 6–10 | Strongly agree | Strongly agree |
| Ric | 66–70 | Obese | Y | Y | Y | N | Y | 16–20 | 15 | Agree | Agree |
| Ron | 51–55 | Normal | Y | Y | Y | Y | N | 16–20 | 16–20 | Undecided | Undecided |
| Sam | 56–60 | Overweight | Y | Y | Y | Y | N | 21–25 | 16–20 | Strongly agree | Strongly agree |

Participants above the horizontal line strongly agreed or agreed that participation in cricket resulted in an increase in current physical activity level and participants below the horizontal line strongly agreed or agreed that participation in cricket resulted in a decrease in current physical activity level.

*Ranges were reported rather than absolute values to assure participants' anonymity.

BMI (body mass index)=categorised with reference to WHO international classification guidelines (normal weight: 18.9–24.9 kg/m², overweight: 25.0–29.9 kg/m², obese: ≥30.0 kg/m²).[20]

Joint pain='do you experience pain, discomfort or have a problem with your: hip(s) or groin, knee(s), ankle(s), spine (back or neck), shoulder(s), elbow(s), wrist(s), finger(s) or hand(s)'.

OA (osteoarthritis)='have you ever been told you have wear and tear, degeneration or osteoarthritis by a doctor?'.

Past injury='have you ever had any injuries leading to more than 4 weeks of reduced participation in exercise, training or sport?'.

Arthroscopic joint surgery=one or more arthroscopic joint surgery.

Joint replacement='have you ever had joint replacement surgery?'.

UK professional seasons=number of seasons playing professional cricket in the UK.

Considering the benefits and risks of cricket, I would do the same again=considering the benefits and risks of my previous participation in cricket, I would do the same again (responses on a five-point Likert scale: strongly agree, agree, undecided, disagree, strongly disagree).

I would recommend cricket to significant others=considering the benefits and risks of my previous participation in cricket, I would recommend this to my children, relatives or close friends (responses on a five-point Likert scale: strongly agree, agree, undecided, disagree, strongly disagree).

Y, yes; N, no; NR, not reported.

*17 years, so yeah they're just things that exist, but no, I'm not that bothered."*

### The ups and downs of professional cricket

Retired cricketers described experiencing mental hardship and psychological challenges during their cricket career. These included coming to terms with experiencing more failure than success and difficulty coping with pressure to perform. Almost all participants believed these experiences benefited other areas of life after retirement. Specifically, they described how attributes required to succeed on the cricket field, including mental toughness, confidence and resilience, equipped them with tools to overcome postretirement life challenges, including managing chronic pain and establishing a successful career outside of cricket.

*Lee: "It was just a constant battle, it was soul destroying at times but out of all that you know; hard work, disappointment, you know, being beaten down by people at times… I think it's given me the mental capacity just to, just get your head down and battle on. I think that's the main thing, you can have bad times in whatever you're doing, and you're going to feel really low, but I think just that mental capacity that you've learnt from cricket, from being a professional sportsmen, to just get back up on your feet and fight another day sort of thing is being part of where I've got to now, you know thanks to that basically."*

Only one individual felt that the mental aspects of professional cricket had a negative impact on life after retirement. Dan attributed difficulty coping with adversity and reduced confidence to having played professional cricket.

*Dan: "I wouldn't say I carry mental scars, but I think it's affected my personality… probably I am less confident having been a professional sportsman than I would have been otherwise, because I think I have been beaten up quite a few times over the years, you know mentally, and it's taken its toll a little bit."*

### Psychological attributes that may enhance QOL
#### Resilience and a positive attitude

Attributes that appeared to enhance QOL in this sample of retired cricketers, included resilience and a positive attitude. Almost all retired cricketers in this study demonstrated characteristics and described experiences suggesting high levels of resilience and a positive attitude regarding pain and musculoskeletal impairment.

*Dom: "I would classify myself to be very, very lucky, probably very, very lucky to come out the other side, with just, at the moment, hip injuries that have been cured by hip operations… it's all been extremely positive, from a life style I've reached, yeah."*

*Ron: "Keep a strong mind and you will have aliments depending on how much sport you have played and you just have to get on with it."*

### Body awareness, acknowledging limitations and self-management

There were other common attributes among the retired cricketers that appeared to have a positive impact on QOL, these included a heightened sense of body awareness and confidence in self-managing physical ailments. Most participants had made positive adaptations and were mindful of their body's capabilities. Participants adhered to self-imposed activity limits and modified their activity choices on a day to day basis to avoid exacerbation of pain or an acute deterioration in physical function.

*Fin: "…you also have a greater and wiser knowledge of what you should, could, and can and can't do. So from a positive point of view you know what you can and can't do and you know what's good for you and your body."*

*Gus: "I like to think that I know what my, what my boundaries are and I can just push them a little bit or to the limit that I think that, that my age will allow me to do… I definitely think that playing cricket made me certainly more aware and being involved in cricket for the amount of time that I had, I certainly have, have more knowledge of what I can and can't do."*

### Fortunate, proud and grateful to have played professional cricket

Reflecting on their past career in cricket, all participants felt fortunate, proud or grateful to have had the opportunity to play professional cricket. All but one individual believed that overall, the positive effects of cricket outweighed the negative, and expressed no regret over playing professional cricket.

*Fin: "No regrets what so ever. It was a great life. I had a great time. There were things that, when you come out of that sporting environment into the real world, if you like, and you realise that then its only things that other people dream of doing, is what you've done. So absolutely no regrets what so ever."*

*Sam: "You just play the hand your dealt and I never think there's any point having any regrets or bearing any grudges, it is what it is and you know I've had a great life out of it. …I've made plenty of rubbish decisions, but I wouldn't change that fact, because all that failure and all that success has made me who I am and I'm quite happy being me now."*

### Quality of life

Irrespective of pain and joint difficulties, most retired cricketers were very satisfied (n=11, 61%) or satisfied (n=6, 33%) with their QOL. No participants expressed that their QOL was impaired due to pain or physical issues. The only individual dissatisfied with his QOL attributed this entirely to current family difficulties.

Interviewer: Overall how satisfied are you with your current quality of life?

*Wes: "110%, cannot believe my luck…I am spoilt rotten, but I mean there's no point in being in a brilliant position, which I am, if you don't appreciate it, I mean I can see lots of people who don't know how lucky they are, well, I am glad to say I do."*

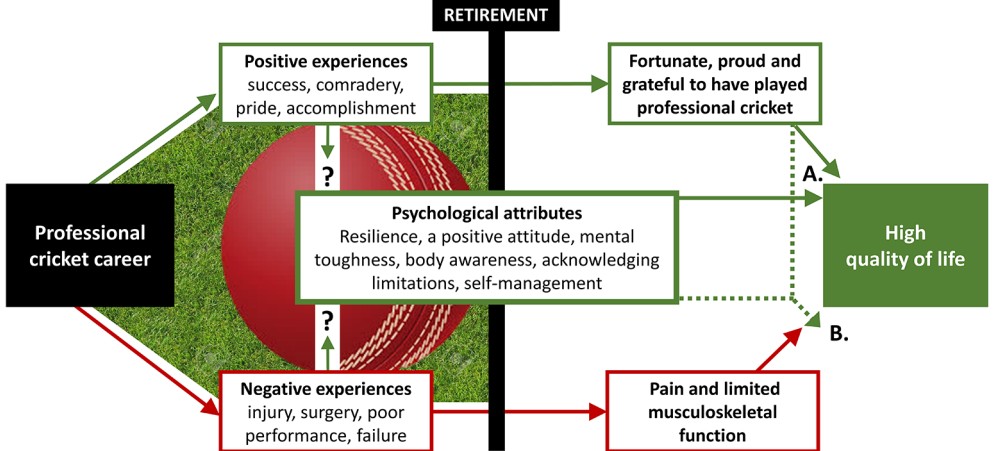

**Figure 1** A conceptual flow diagram summarising main findings and potential interactions between key themes. Green arrows and boxes represent factors with potential to *positively* impact quality of life. Red arrows and boxes represent factors with potential to *negatively* impact quality of life. (A) Reflecting on positive experiences in cricket and psychological attributes may positively impact quality of life following retirement from professional cricket. (B) Reflecting on positive experiences in cricket and psychological attributes may reduce the impact of pain, osteoarthritis and physical limitations on quality of life after retirement from professional cricket. Together, A and B provide one potential explanation for high quality of life despite a high prevalence of pain, osteoarthritis and physical limitations in this sample of retired professional cricketers. Question marks denote uncertainty surrounding the nature of the relationship between cricket-related experiences and psychological attributes common in successful cricketers, since individuals may possess these attributes prior to cricket participation.

Sam: *"I am absolutely delighted with it, I love it. I'm in a really good place mentally… things couldn't be better."*

Leo: *"Oh brilliant, I'm just very, very lucky, absolutely brilliant."*

Ken: *"I am very happy with it actually, and have been for a number of years."*

Joe: *"Good, yeah I'm very happy."*

Guy: *"Oh on a scale, I would say 8/9 out of 10; I think I am very satisfied."*

Gus: *"Well I am very happy, very happy and pleased with my quality of life."*

Dom: *"Very happy…very happy with where I am physically and mentally in what I do and what I want to do, yes."*

Ned: *"Oh generally, you know, very happy really, I think you know keeping things in perspective of, you know of how I am, my age, you know, what I've got, you know I can't complain really. I am really pretty pleased with the way things are."*

### Summary of key themes
Figure 1 provides a conceptual flow diagram summarising the main findings and providing an overview of how the key themes may interact to make a positive contribution to QOL after retirement.

### DISCUSSION
Current pain and physical limitations were common in this group of retired professional cricketers. Despite this, almost all retired cricketers were satisfied or very satisfied with their QOL. One potential explanation for this counterintuitive finding is the common psychological attributes that these individuals may possess, including

resilience, a positive attitude, heightened body awareness, acknowledgement of physical limitations and an ability to modify activity choices to effectively cope with fluctuations in pain and musculoskeletal function. Reflecting on their past cricket career, participants felt very grateful and fortunate to have played professional cricket, and all but one felt the positive impacts of a career in cricket outweighed any negatives. This feeling of pride and accomplishment has further potential to positively influence QOL in this group of retired cricketers, irrespective of pain and physical impairment.

### QOL and flourishing
We found that all but one retired cricketer was satisfied with his QOL. Flourishing is a concept that represents an optimal QOL where an individual excels emotionally, psychologically and socially. In 2013, 20% of a representative sample of the UK general population were found to be 'flourishing' (possessing 9 of 10 signs of positive well-being including resilience, optimism, competence, self-esteem and meaning).[21] Retired cricketers who took part in the interviews exhibited many attributes consistent with the definition of flourishing. It is possible that a greater number of retired-cricketers 'flourish' in later life compared with the general population despite a higher prevalence of pain and osteoarthritis, although further research is necessary to explore this possibility.

Retired professional footballers with osteoarthritis reported worse health-related QOL compared with retired footballers without osteoarthritis.[22] However, most conventional QOL outcome measures (including the EQ-5D used in the study of retired footballers[22]) are influenced by pain and physical impairment, where the presence of pain and physical disability will result in a

reduced QOL score, irrespective of the impact of pain or disability on the individual.[23] Evaluation of the psychometric properties of QOL measures in retired sporting populations is warranted. The Flourishing Scale[24] or a QOL measure not driven by pain and function (such as the WHO-QOL[25]) may be preferable for use in retired sporting populations.

### Resilience and associated positive psychological attributes

Resilience refers to responding to adversity with positive adaptation, and in doing so, restoring an internal sense of emotional and psychological balance.[26 27] Retired cricketers in the present study shared resilient attributes; this was highlighted by their ability to maintain a positive attitude and adapt activity choices to effectively cope with fluctuations in pain and physical function. Prior research has found that athletes possess greater resilience and optimism than non-athletes[28] and successful cricketers are likely to be more resilient than less successful cricketers.[10 12] A high level of resilience is associated with increased QOL in a wide variety of disease-specific and chronic pain populations.[29–36] Resilient individuals also demonstrate more adaptive coping styles and better musculoskeletal pain adjustment.[37 38] Considering sporting injury is a risk factor for osteoarthritis, interventions to build resilience within a sporting setting[39] may have benefits that extend beyond success on the sporting field, with potential to enhance QOL in later life.

Despite these benefits, who is likely to become resilient and why is poorly understood.[40] The relationship between resilience and professional cricket is likely multifaceted. Exposure to the demands of cricket from a young age, including repetitive failure, disappointment, stress and adversity may facilitate the development of resilience. On the other hand, resilience may be influenced by a range of non-cricket-related factors, and resilient individuals may be most likely to reach professional status resulting in a high level of resilience in professional cricketers. It is also possible that adversities faced during transition from sport and after retirement contributed to the high level of resilience in this group of retired cricketers.

### Cricket reflection

Accomplishment refers to success, achievement or mastery at the highest possible level[41] and has been recognised as an important component of well-being.[42] Considering all retired cricketers in this study were fortunate, proud or grateful to have played professional cricket, it is possible that this accomplishment further contributed to the high QOL described by these individuals. A retired athlete's sense of accomplishment and fulfilment through past participation in sport is too often overlooked in research, despite potential to positively influence QOL.

### Study strengths and potential limitations

It is not known whether cricket participation contributed to the development of positive psychological attributes or whether these largely pre-existed prior to cricket participation or developed after retirement. Prospective longitudinal studies would be required to answer this question. Although participants were asked about their degree of satisfaction with their overall QOL, we did not explore personal beliefs about the definition of QOL. Consequently, the question pertaining to QOL could have meant different things to different participants. Due to purposive recruitment, individuals reporting uncertainty regarding the impact of cricket on their physical activity level were not invited into the study, reducing the generalisability of results. It is also possible that the study was subjected to selection bias, whereby retired cricketers with specific characteristics were more likely to desire participation in a qualitative interview study. Repeat interviews were not performed and transcripts were not returned to participants for correction or comment; these procedures could have provided additional information and insights beyond those gained through the initial interviews. A strength of this study was the strong themes and consistent patterns that emerged through the interviews resulting in saturation of data at 14 interviews, confirmed by the final four interviews that enabled further complimentary exploration into ideas and important themes. The interviewer was a physiotherapist with experience in building rapport with individuals and capturing issues of importance to an individual. Rapport was further strengthened by the interviewer's familiarity with cricket as a physiotherapist and cricket participant. These factors facilitated deep reflection and open, personal insights from retired cricketers that enriched the findings of this study.

### CONCLUSION

Irrespective of chronic joint pain and physical limitations, all but one retired cricketer in this study was satisfied or very satisfied with their QOL. Retired cricketers shared common psychological attributes including resilience and a positive attitude that may partly explain the small impact pain and physical limitations appeared to have on the QOL of these retired cricketers. Additionally, most retired cricketers had a high sense of body awareness and executed self-control to adapt activity choices and accept physical limitations. Participants felt fortunate and grateful to have played professional cricket and expressed that the benefits of cricket participation outweigh any negative impacts. This study highlights fruitful areas for future research including (1) flourishing in retired athletes despite pain and functional limitations, (2) the potential for positive psychological attributes to develop through sport participation and impact QOL after retirement and (3) the effect of past accomplishment in sport on QOL across the lifespan. A better understanding of the relationship between sport participation and QOL could assist in promoting physical activity and facilitate strategies to enhance the positive impact of sport on QOL.

**Twitter** @stephfilbay

**Acknowledgements** We would like to thank the participants for volunteering their time to take part in this study. We would also like to thank Angus Porter and

the Professional Cricketers' Association (PCA) for assisting with recruitment and questionnaire development for the larger cross-sectional study from which study participants were purposively recruited.

**Contributors** SRF, FB, NP and NKA conceived and designed this qualitative study. SRF and MEJ recruited participants and extracted data form the cross-sectional cohort. SRF performed all interviews. SRF and FB participated in the analysis. SRF drafted the first version of the manuscript. All authors contributed in revising the manuscript and gave their final approval of the submitted version.

**Patient consent** Obtained.

**Ethics approval** Medical Sciences Inter-divisional Research Ethics Committee (IDREC), University of Oxford.

**Provenance and peer review** Not commissioned; externally peer reviewed.

**Data sharing statement** To view interview transcripts or additional participant quotes, please contact the corresponding author.

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
