## [Reviewer comments · BMJ Open]

ARTICLE DETAILS

TITLE (PROVISIONAL)	Common attributes in retired professional cricketers that may enhance or hinder quality of life after retirement: A qualitative study
AUTHORS	Filbay, Stephanie; Bishop, Felicity; Peirce, Nicholas; Jones, Mary; Arden, Nigel

VERSION 1 - REVIEW

REVIEWER	Gustavo Gonçalves Arliani Universidade Federal de São Paulo/ Brazil
REVIEW RETURNED	16-Mar-2017

GENERAL COMMENTS	The study is interesting and has a current issue. Abstract is appropriate Introduction OK Methods - Why the authors did not also use QOL subjective questionnaires? The sample is short and an author interview is a vies. Some important papers in that theme were not cited by the authors: - Arliani GG, Astur DC, Yamada RKF, et al. Early osteoarthritis and reduced quality of life after retirement in former professional soccer players. Clinics. 2014;69(9):589-594. doi:10.6061/clinics/2014(09)03.- Turner AP, Barlow JH, Heathcote-Elliott C. Long term health impact of playing professional football in the United Kingdom. Br J Sports Med. 2000;34(5):332-6.- Arliani GG, Lara PS, Astur DC, Cohen M, Gonçalves JPP, Ferretti M. Impact of sports on health of former professional soccer players in Brazil. Acta Ortopedica Brasileira. 2014;22(4):188-190. doi:10.1590/1413-78522014220400954.
--

REVIEWER	Dr Janine Gray Cricket South Africa/University of Cape Town South Africa
REVIEW RETURNED	30-Mar-2017

GENERAL COMMENTS	Quality of life following retirement from professional cricket: retired players' perspective. Thank you for asking me to review this paper. I feel there is a lot of merit to this paper; it highlights novel findings not previously described and uses sound methodology to collect the data. I have made a few minor comments below. Introduction While I feel the introduction includes all the necessary information to
---

set the scene for the study I feel that a number of themes are discussed interchangeably in a single paragraph and that these could be grouped better to make the point. The 2nd paragraph described the psychological attributes of cricketers during play, the impact of retirement on psychology, QOL to OA and a cricketer's ability to deal with pain in retirement all in 1 paragraph. The flow of the introduction would be better if concepts were explained more fully and grouped together.

Page 6, line 37; The primary aim is reported to explore physical activity after retirement. However, this would appear to be the primary aim of the larger study not this one. Please correct this to present the aims of this study. The main question of this study therefore appears to be twofold:

1. The influence of musculoskeletal pain in retired cricketers on their QOL.

2. The psychological attributes of retired cricketers that may influence their retirement quality of life.

Methodology

Page 7, line 36: You indicate that your participants were selected from a larger sample based on their answer to 2 questions on physical activity. You need to indicate that you chose equal numbers of participants from each group (this is only clarified later in the methodology).

The inductive thematic approach used in this study has been well described.

Results

Results are clearly presented and well supported by appropriate quotes. It should be highlighted that the results are a combination of qualitative and quantitative analysis. The interview findings have been used to explain the findings of high musculoskeletal injury and pain and the high quality of life experienced by cricketers.

Page 11, line 3: Please could clarify if this past history of injury was 'ever' or pre/post retirement.

Page 12, line 42L "Language used throughout the ..." I feel the word language does not adequately explain how the investigators came to make the conclusions they did. This may be pedantic but I feel this is better explained by the process of identifying themes and whether these themes are reflected positively or negatively.

Page 13, line 226: Does the 21 test matches described by Sam not make him identifiable.

Page 14, line 53: Figure 1: This figure does not accurately describe what you have described in your results. As you point out in the discussion it is unclear whether a resilient person becomes a successful cricketer or the sport helps an individual develop resilience. Your figure suggests that the positive and negative experiences develop resilience which you cannot conclude from your study. No examples are given for positive experience like have been included for the negative experience. Positive experiences produce emotions (fortunate, proud etc) while the negative experiences produced a physical experience (pain and function). This doesn't make sense. I feel this figure needs to be revised.

Discussion

Page 15, line 57: "...who took part in this study described many attributes.....". Demonstrated indicates a more measurable assessment of the attribute.

Page 16, line 3: "...greater proportion of retired, professional cricketers...".

Page 16, line 10: "QOL outcome measures are driven influenced by pain and physical impairment...". "Driven by" indicates they are the sole determinants of QOL. Success, money, etc would surely

	influence QOL too. Page 16, line 34: "...the present study frequently described shared resilient attributes, this which was highlighted...". This sentence is very long. Please shorten it. Conclusion Appropriate.
--	---

VERSION 1 – AUTHOR RESPONSE

Reviewer: 1 Comments

1.1 Reviewer comment

The study is interesting and has a current issue.

Abstract is appropriate

Introduction OK

Methods - Why the authors did not also use QOL subjective questionnaires?

1.1 Author response

We agree that QOL questionnaires would have provided additional information that could have been a useful supplement to the qualitative interviews. However, assessing QOL was not an aim of the larger cross-sectional study from which participants were recruited and consequently validated measures of QOL were not included in the cross-sectional questionnaire design.

1.1 Author action

No action taken.

1.2 Reviewer comment

The sample is short and an author interview is a vies.

1.2 Author response

We believe the decision to perform 18 interviews was appropriate. The decision to cease recruitment at 18 interviews was supported by data saturation being achieved by the 14th interview. This was further confirmed by an additional four interviews that were carried out after data saturation, where no new themes emerged.

1.2 Author action

No action taken.

1.3 Reviewer comment

Some important papers in that theme were not cited by the authors:

- Arliani GG, Astur DC, Yamada RKF, et al. Early osteoarthritis and reduced quality of life after retirement in former professional soccer players. *Clinics*. 2014;69(9):589-594.
doi:10.6061/clinics/2014(09)03.

- Turner AP, Barlow JH, Heathcote-Elliott C. Long term health impact of playing professional football in the United Kingdom. *Br J Sports Med*. 2000;34(5):332–6.

- Arliani GG, Lara PS, Astur DC, Cohen M, Gonçalves JPP, Ferretti M. Impact of sports on health of former professional soccer players in Brazil. *Acta Ortopedica Brasileira*. 2014;22(4):188-190.
doi:10.1590/1413-78522014220400954.

1.3 Author response

Thank you for these suggestions. Although we cannot cite all relevant papers in this area, we have edited the introduction to reference Arliani et al 2014, and the discussion to reference Turner et al 2000, since this directly aligns with a key point raised in the discussion.

1.3 Author action

p5 Lines 6-8

After retirement, athletes are at increased risk of developing osteoarthritis, predominately due to a history of sport-related injury, and this can result in pain and activity limitations.³⁻⁷

p16. Lines 295-303:

Retired professional footballers with osteoarthritis reported worse health-related QOL compared with retired footballers without osteoarthritis.²² However, most conventional QOL outcome measures (including the EQ-5D used in the study of retired footballers²²) are influenced by pain and physical impairment, where the presence of pain and physical disability will result in a reduced QOL score, irrespective of the impact of pain or disability upon the individual.²³

Reviewer: 2 comments

Thank you for asking me to review this paper. I feel there is a lot of merit to this paper; it highlights novel findings not previously described and uses sound methodology to collect the data. I have made a few minor comments below.

2.1 Reviewer comment

Introduction

While I feel the introduction includes all the necessary information to set the scene for the study I feel that a number of themes are discussed interchangeably in a single paragraph and that these could be grouped better to make the point. The 2nd paragraph described the psychological attributes of cricketers during play, the impact of retirement on psychology, QOL to OA and a cricketer's ability to deal with pain in retirement all in 1 paragraph. The flow of the introduction would be better if concepts were explained more fully and grouped together.

2.1 Author response

Thank you for your constructive feedback. We have reformatted the introduction to improve the flow, as suggested.

2.1 Author action

p5-7, Lines 10-44:

Introduction reformatted, as per track changes.

2.2 Reviewer comment

Page 6, line 37; The primary aim is reported to explore physical activity after retirement. However, this would appear to be the primary aim of the larger study not this one. Please correct this to present the aims of this study. The main question of this study therefore appears to be twofold:

1. The influence of musculoskeletal pain in retired cricketers on their QOL.

2. The psychological attributes of retired cricketers that may influence their retirement quality of life.

2.2 Author response

We see how presenting the study aim in this way may create confusion. We do however, believe it is important to accurately report both the a priori aim of the overall qualitative study, and the specific aim of this paper. We have revised the wording and hope this improves clarity whilst still making a distinction between the overall study aim, and specific aim of this paper.

2.2 Author action

We have changed the wording to improve clarity:

p6-7, Lines: 47-49: The a priori aim of this qualitative study was to explore physical activity after retirement from professional cricket. However, the study also captured participants' broader perspectives regarding QOL and prominent themes linking QOL to professional cricket emerged during the initial inductive analysis. This paper focuses on these emergent themes, which were explored while considering two sensitising questions: How do retired cricketers perceive and experience musculoskeletal pain and function in daily life? Are there any psychological attributes that might enhance or hinder retired cricketers' QOL? A complementary manuscript will address physical activity after retirement from professional cricket.

2.3 Reviewer comment

Methodology

Page 7, line 36: You indicate that your participants were selected from a larger sample based on their answer to 2 questions on physical activity. You need to indicate that you chose equal numbers of participants from each group (this is only clarified later in the methodology).

The inductive thematic approach used in this study has been well described.

2.3 Author response

Thank you for pointing this out, we have now included an additional sentence highlighting this.

2.3 Author action

p8 Line 73-74:

We aimed to include an equal number of participants from each of these groups.

2.4 Reviewer comment

Results

Results are clearly presented and well supported by appropriate quotes. It should be highlighted that the results are a combination of qualitative and quantitative analysis. The interview findings have been used to explain the findings of high musculoskeletal injury and pain and the high quality of life experienced by cricketers.

2.4 Author response

Although pain and osteoarthritis were assessed quantitatively in the larger cross-sectional study via questionnaire, this data did not contribute to the qualitative analysis. However, the cross-sectional data was used to describe the sample characteristics in Table 1. We realise this may not be clear, and have subsequently clarified this distinction in the methods section.

2.4 Author action

The following paragraph was added to the methods section:

p10-11, Lines 138-143:

Following qualitative analysis, responses to relevant questions collected as part of the cross-sectional

study were used to describe the study sample (including osteoarthritis status, age, previous surgery and length of professional cricket career). Questionnaire data was cross-checked for accuracy with participants' narratives and if disparities were present (e.g. a participant reported no current joint pain on questionnaire but described experiencing current joint pain during the interview), participant narratives were considered more reliable.

2.5 Reviewer comment

Page 11, line 3: Please could clarify if this past history of injury was 'ever' or pre/post retirement.

2.5 Author response

This has now been clarified.

2.5 Author action

p11 Lines 155-157:

More than two thirds (n=14, 78%) reported having ever had an injury that resulted in more than one month of reduced participation in exercise, training or sport.

2.6 Reviewer comment

Page 12, line 42L "Language used throughout the ..." I feel the word language does not adequately explain how the investigators came to make the conclusions they did. This may be pedantic but I feel this is better explained by the process of identifying themes and whether these themes are reflected positively or negatively.

2.6 Author response

Indeed it was not merely the language used that demonstrated a high level of resilience amongst participants. To clarify, we have removed the reference to participant language.

2.6 Author action

p12, Lines 190-192

Attributes that appeared to enhance QOL in this sample of retired cricketers, included resilience and a positive attitude. Language used throughout the interviews highlighted that Almost all retired-cricketers in this study were resilient and had a positive attitude regarding pain and musculoskeletal impairment.

2.7 Reviewers comment

Page 13, line 226: Does the 21 test matches described by Sam not make him identifiable.

2.7 Authors response

We have decided to remove the start of the quote where reference to the number of test matches played is made.

2.7 Authors action

p14 Line 242-244

'You just play the hand your dealt and I never think there's any point having any regrets or bearing any grudges, it is what it is and you know I've had a great life out of it. ...I've made plenty of rubbish decisions, but I wouldn't change that fact, because all that failure and all that success has made me who I am and I'm quite happy being me now.'

2.8 Reviewer comment

- a. Page 14, line 53: Figure 1: This figure does not accurately describe what you have described in your results. As you point out in the discussion it is unclear whether a resilient person becomes a successful cricketer or the sport helps an individual develop resilience. Your figure suggests that the positive and negative experiences develop resilience which you cannot conclude from your study.
- b. No examples are given for positive experience like have been included for the negative experience.
- c. Positive experiences produce emotions (fortunate, proud etc) while the negative experiences produced a physical experience (pain and function). This doesn't make sense. I feel this figure needs to be revised.

2.8 Author response

- a. Thank you for this feedback. You are quite right, we cannot insinuate a causal relationship between positive/negative experiences while playing cricket, and psychological attributes such as resilience in professional cricketers. We only intended to highlight 'potential' interactions between key themes as stated in the title. However, to ensure this is not misleading we have revised the figure to include question marks and have included a description of what this denotes in the figure legend.
- b. We have now included examples of positive experiences to improve consistency.
- c. With the inclusion of the arrow and question mark between experience and psychological attributes, the negative experience may result in physical (pain and limited function), behavioural (body awareness, self-management) and emotional/psychological (eg, resilience, mental toughness) impacts.

If the arrow/question mark between negative experiences and psychological attributes was removed, than you are quite right that this would not make sense since negative experiences would impact only physical factors without having psychological consequences.

2.8 Author action

p25 Figure 1.

- a. Question marks have been added to Figure 1 and a description of what this denotes has been added to the figure legend:
'Question marks denote uncertainty surrounding the nature of the relationship between cricket-related experiences and psychological attributes common in successful cricketers, since individuals may possess these attributes prior to cricket participation'
- b. The examples given for 'positive experiences' are as follows: success, comradery, pride, accomplishment
- c. No changes made since this has been addressed by adding question marks (see 2.8 a.)

2.9 Reviewer comment

Discussion

Page 15, line 57: "...who took party in this study described many attributes.....". Demonstrated indicates a more measurable assessment of the attribute.

2.9 Author response

Thank you for this suggestion. We agree that demonstrated may not be an ideal word to use here. However, participants did not 'describe' these attributes, rather, they 'exhibited' them. Thus, we have replaced the word 'demonstrated' with 'exhibited.'

2.9 Author action

p16 line 306. 'Demonstrated' has been replaced with 'exhibited.'

2.10 Reviewer comment

Page 16, line 3: "...greater proportion of retired, professional cricketers...".

2.10 Author response

We have replaced 'proportion' with 'number'

2.10 Author action

p16 line 307: 'It is possible that a greater number of retired-cricketers 'flourish' in later life compared with the general population'

2.11 Reviewer comment

Page 16, line 10: "QOL outcome measures are driven influenced by pain and physical impairment...". "Driven by" indicates they are the sole determinants of QOL. Success, money, etc would surely influence QOL too.

2.11 Author response

The majority of health related QOL measures, including the EQ-5D and SF-36, are largely influenced by the presence of pain or physical disability. Yet, they do not quantify the contribution of pain or physical disability to an individual's QOL. Thus, if a person is physically disabled, the overall QOL score will be impaired. Nevertheless, we agree that 'driven' may be somewhat misleading, and 'influenced' would be more appropriate here.

2.11 Author action

p16 – lines 311-314

However, most conventional QOL outcome measures (including the EQ-5D used in the study of retired footballers²²) are influenced by pain and physical impairment, where the presence of pain and physical disability will result in a reduced QOL score, irrespective of the impact of pain or disability upon the individual.²³

2.12 Reviewer comment

Page 16, line 34: "...the present study frequently described shared resilient attributes, this which was highlighted....". This sentence is very long. Please shorten it.

Conclusion

Appropriate.

2.12 Author response

This sentence has been shortened.

2.12 Author action

p17 Line 352:

Retired-cricketers in the present study shared resilient attributes, this was highlighted by their ability to maintain a positive attitude and adapt activity choices to effectively cope with fluctuations in pain and physical function.

VERSION 2 – REVIEW

REVIEWER	D Janine Gray Cricket South Africa/University of Cape Town South Africa
REVIEW RETURNED	02-May-2017

GENERAL COMMENTS	I have reviewed the changes made by the authors in response to my previous review. I am satisfied with the changes in the text. The changes to figure are sufficient although I am still not clear on how the negative experiences affect the pain and limited function on a continuum to the QOL. However, if other reviewers are satisfied I am happy to accept changes. I am happy that this is now publishable. 2 minor comments include: Page 12, line 192: "...in this study were resilient...". As resilience wasn't measured directly it would be more accurate to say they describe characteristics akin to resilience. Page 15, line 276: "... these individuals to possess,". Again, I think it would be more accurate to say these characteristics were described.
--

VERSION 2 – AUTHOR RESPONSE

Reviewer(s)' Comments to Author:

Reviewer: 2

Dr Janine Gray

Cricket South Africa/University of Cape Town, South Africa

Please state any competing interests or state 'None declared': None declared

2.1 Reviewer comment

Please leave your comments for the authors below

I have reviewed the changes made by the authors in response to my previous review. I am satisfied with the changes in the text. The changes to figure are sufficient although I am still not clear on how the negative experiences affect the pain and limited function on a continuum to the QOL. However, if other reviewers are satisfied I am happy to accept changes. I am happy that this is now publishable.

2.1 Author response

We would like to thank reviewer 2 for taking the time to critically appraise the manuscript and provide valuable comments and feedback. Your attention to detail is greatly appreciated.

In Figure 1, injury and surgery are described as 'negative experiences'. Injury and surgery are risk factors for developing chronic pain and osteoarthritis in later life, which a majority of participants reported and attributed to having played professional cricket. As described in the discussion, chronic pain and osteoarthritis have been associated with worse quality of life in other samples and general population groups. What is proposed in this figure, is that psychological attributes (including resilience, acknowledging limitations and self-management), as well as pride surrounding past accomplishments in cricket, may in part, negate the negative impacts of pain and osteoarthritis upon quality of life.

2.1 Author action

We have made small edits to the wording of Figure 1 legend to provide further clarity:

p. 29

Green arrows and boxes represent factors with potential to positively impact quality of life;

Red arrows and boxes represent factors with potential to negatively impact quality of life;

A. Reflecting upon positive experiences in cricket and psychological attributes may positively impact

quality of life following retirement from professional cricket;

B. Reflecting upon positive experiences in cricket and psychological attributes may reduce the impact of pain, osteoarthritis and physical limitations on quality of life after retirement from professional cricket;

Together, A. and B. provide one potential explanation for high quality of life despite a high prevalence of pain, osteoarthritis and physical limitations in this sample of retired professional cricketers;

Question marks denote uncertainty surrounding the nature of the relationship between cricket-related experiences and psychological attributes common in successful cricketers, since individuals may possess these attributes prior to cricket participation

2.2 Reviewer comment

2 minor comments include:

Page 12, line 192: "...in this study were resilient...". As resilience wasn't measured directly it would be more accurate to say they describe characteristics akin to resilience.

Page 15, line276: "... these individuals to possess,". Again, I think it would be more accurate to say these characteristics were described.

2.2 Author response

Thank you for drawing attention to this, we have made the following amendments:

2.2 Author action

p13. "Almost all retired-cricketers in this study demonstrated characteristics and described experiences suggesting high levels of resilience and a positive attitude regarding pain and musculoskeletal impairment."

p15. "...psychological attributes that these individuals may possess, including resilience, a positive attitude, heightened body awareness.."